# Beyond Labels: Empowering Human Annotators with Natural Language Explanations through a Novel Active-Learning Architecture

**Bingsheng Yao**
Rensselaer Polytechnic Institute

**Ishan Jindal**
IBM Research

**Lucian Popa**
IBM Research

**Yannis Katsis**
IBM Research

**Sayan Ghosh**
UNC Chapel Hill

**Lihong He**
IBM Research

**Yuxuan Lu**
Northeastern University

**Shashank Srivastava**
UNC Chapel Hill

**Yunyao Li**[†]
Apple

**James Hendler**
Rensselaer Polytechnic Institute

**Dakuo Wang** [*]
Northeastern University

## Abstract

Real-world domain experts (e.g., doctors) rarely annotate only a decision label in their day-to-day workflow without providing explanations. Yet, existing low-resource learning techniques, such as Active Learning (AL), that aim to support human annotators mostly focus on the **label** while neglecting the **natural language explanation** of a data point. This work proposes a novel AL architecture to support experts' real-world need for label and explanation annotations in low-resource scenarios. Our AL architecture leverages **an explanation-generation model** to produce explanations guided by human explanations, **a prediction model** that utilizes generated explanations toward prediction faithfully, and **a novel data diversity-based AL sampling strategy** that benefits from the explanation annotations. Automated and human evaluations demonstrate the effectiveness of incorporating explanations into AL sampling and the improved human annotation efficiency and trustworthiness with our AL architecture. Additional ablation studies illustrate the potential of our AL architecture for transfer learning, generalizability, and integration with large language models (LLMs). While LLMs exhibit exceptional explanation-generation capabilities for relatively simple tasks, their effectiveness in complex real-world tasks warrants further in-depth study.

## 1 Introduction

State-of-the-art (SoTA) language models (Devlin et al., 2019; Radford et al., 2019; Winata et al.,

---
[†]d.wang@northeastern.edu Corresponding Author. Work done while Yunyao was at IBM Research.

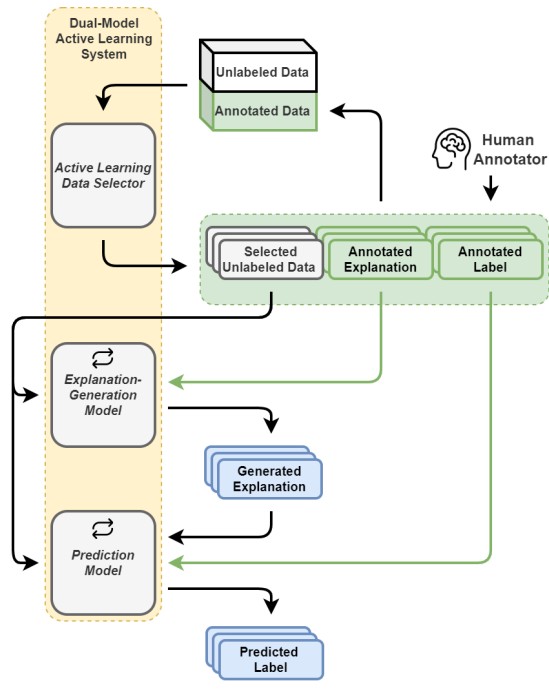

Figure 1: Our dual-model AL system architecture at every iteration: 1) the AL data selector chooses a few unlabeled examples; 2) human annotators provide an explanation and label for each data instance; 3) the annotated explanations are used to finetune the explanation-generation model; 4) the annotated labels and generated explanations are used to finetune the prediction model. Then, humans can review the predicted labels and generated explanations for unlabeled data and start the next iteration. Green arrows indicate the training target.

2021) demonstrate astonishing performance on various NLP tasks, including Question Answering (QA) and Question Generation (QG) (Rajpurkar et al., 2016; Duan et al., 2017; Kočiský et al., 2018; Yao et al., 2022), Natural Language Inference (NLI) (Bowman et al., 2015; Wang et al.,

2018), etc. Despite the superior generative capabilities, the lack of faithful explainability within these "black boxes" may lead to mistrust of their predictions (Lipton, 2018), where humans, on the other hand, can develop intermediate rationales to facilitate the decision-making process.

The lack of explainability and untrustworthiness of models is magnified in the real world (Drozdal et al., 2020), where domain experts rarely only annotate a decision label in their daily workflow without providing explanations (i.e., clinical diagnoses by clinicians) (Zhang et al., 2023), and humans need explanations to understand and trust model predictions (Zhang et al., 2021). Therefore, a few approaches were proposed to retrospectively analyze the probability distribution within the model or ask models to generate explanations along with predictions (Ribeiro et al., 2016; Lundberg and Lee, 2017; Yu et al., 2019; Rajagopal et al., 2021; Chen et al., 2021), despite, the former is still very difficult for laymen to understand while the latter explanations are not faithful toward predictions.

As researchers looked into the quality (Carton et al., 2020; Yao et al., 2023) of human-annotated natural language explanations (Camburu et al., 2018; Rajani et al., 2019; Aggarwal et al., 2021), they discovered numerous issues in existing datasets (Geva et al., 2019; Chmielewski and Kucker, 2020; Narang et al., 2020; Sun et al., 2022), such that the human annotations are of low quality and significant inconsistency. Furthermore, the ever-increasing costs in terms of labor, finances, and time for large-scale, high-quality data annotations remain a persistent challenge for the research community. This challenge has given rise to various methodologies to reduce reliance on human annotations, such as Active Learning (AL) (Settles, 2009). AL is a human-in-the-loop framework that utilizes AL sampling strategies to iteratively select a small number of representative examples, request oracle annotations, and subsequently fine-tune the model using the annotated data. However, prior AL works predominantly focus on labels and overlook the fact that real-world scenarios often need both labels and natural language explanations.

In this work, we propose a dual-model AL architecture for human annotation of labels and explanations, drawing inspiration from the human decision-making process. Our system consists of:

1) An explanation-generation model guided by human-provided explanations

2) A prediction model that accepts the data content and the generated explanations for prediction.

We integrate AL to reduce human annotation efforts and establish human trustworthiness by actively engaging humans in the training process. We design a novel data diversity-based AL sampling strategy to select the most representative examples by exploiting the explanation annotations, which is analogous to the prevalent core-set (Sener and Savarese, 2017) strategy. Our AL architecture aims to support low-resource model predictions and AI trustworthiness by explicitly generating natural language explanations. Specifically, we request label and free-form explanation annotations for a very limited number of examples (e.g., 3 or 10) selected by our AL sampling strategy at every AL iteration. Subsequently, the generated explanations serve as input for the final prediction, demonstrating the potential for these explanations to support the model's predictions faithfully.

We conduct two AL simulations with different amounts of samplings and iterations on a large-scale NLI dataset with human-annotated explanations to justify incorporating explanations in AL data selection can consistently outperform random, traditional data diversity-based, and model probability-based sampling strategies. We make the code publically available[1].

A human evaluation of perceived validity, explainability, and preference of the generated explanations among our system, a SoTA explanation-generation system, and human-annotated explanations shows that, despite human explanations being ranked highest, explanations generated by our system are preferred over the SOTA system. Additionally, we conduct three ablation studies to explore the capability and potential of our proposed AL architecture in transfer learning, generalizability, and incorporating large language models (LLMs) for explanation generation to further reduce human efforts. LLMs demonstrate exceptional explanation-generation capabilities on relatively simple tasks. However, their effectiveness in handling complex real-world tasks warrants in-depth study.

## 2 Related Work

### 2.1 Datasets with Natural Language Explanations

Wiegreffe and Marasovic (2021) conducted a com-

---

[1]https://github.com/neuhai/
explanation-enriched-active-learning

prehensive review of 65 datasets with explanations and provided a 3-class taxonomy: highlights, free-text, and structured. Among the large-scale datasets with free-text explanations, **e-SNLI** (Camburu et al., 2018) is a prominent one, which extended the Stanford Natural Language Inference (SNLI) corpus (Bowman et al., 2015), a classification task to determine the inference relation between two textual contexts (premise and hypothesis): entailment, contradiction, or neutral. The e-SNLI dataset (examples are shown in Appendix A) contains human-annotated free-form explanations for $549, 367$ examples in train, $9, 842$ in validation, and $9, 824$ in test split.

Another popular group of datasets extended the Commonsense QA (CQA v1.0 and v1.11 versions) datasets (Talmor et al., 2019), including two variants of Cos-E dataset (**CoS-E v1.0** and **CoS-E v1.11** (Rajani et al., 2019)) and the ECQA (Aggarwal et al., 2021) dataset. Many recent works (Narang et al., 2020; Sun et al., 2022) have found explanations in CoS-E to be noisy and low-quality, and thus, Aggarwal et al. (2021) carefully designed and followed the explanation annotation protocols to created **ECQA**, which is of higher quality compared with CoS-E.

In this paper, we leverage the e-SNLI dataset as the benchmark dataset for our AL simulation experiment because 1) the classification task is popular and representative, 2) the massive data size ensures data diversity, and 3) explanations for a classification task may provide more effective help compared to CQA task where training and testing data may be unrelated. We additionally conduct an ablation study on the ECQA dataset to explore the generalizability of our proposed AL architecture.

## 2.2 Active Learning for Data Annotation

Owning to the paucity of high-quality, large-scale benchmarks for a long tail of NLP tasks, learning better methods for low-resource learning is acquiring more attention, such as Active Learning (AL) (Sharma et al., 2015; Shen et al., 2017; Ash et al., 2019; Teso and Kersting, 2019; Kasai et al., 2019; Zhang et al., 2022). AL iteratively 1) selects samples from the unlabeled data pool (based on AL sampling strategies) and queries their annotation from human annotators, 2) fine-tunes the underlying model with newly annotated data, and 3) evaluates model performance.

A few AL surveys (Settles, 2009; Olsson, 2009; Fu et al., 2013; Schröder and Niekler, 2020; Ren et al., 2021) of sampling strategies provide two high-level selection concepts: data diversity and model probability. We propose a novel **data diversity-based** strategy that leverages human-annotated explanations to select data. Our data selector shares a similar concept with the established data-based clustering strategies (Xu et al., 2003; Nguyen and Smeulders, 2004) and core-set (Sener and Savarese, 2017) that aim to select the most representative data while maximizing diversity. Compared with model probability-based strategies, data diversity-based ones are model-agnostic and need much less computing resources, whereas the former requires inference on unlabeled examples to calculate probability.

In addition to AL, Marasovic et al. (2022) introduces a few-shot self-rationalization setting that asks a model to generate free-form explanations and the labels simultaneously. Similarly, Bhat et al. (2021) proposes a multi-task self-teaching framework with only 100 train data per category, and Bragg et al. (2021) provides guidance on unifying evaluation for few-shot settings.

## 2.3 Natural Language Explanation Generation

Different approaches have been explored to enhance the model's explainability by asking them to generate natural language explanations. Some of them (Talmor et al., 2020; Tafjord et al., 2021; Latcinnik and Berant, 2020) propose systems to generate text explanations for specific tasks. Dalvi et al. (2022) propose a 3-fold reasoning system that generates a reasoning chain and asks users for correction. Other recent works (Paranjape et al., 2021; Liu et al., 2022; Chen et al., 2022) explore different prompt-based approaches to generate additional information for the task and examine the robustness and validity. We believe that our dual-model system provides and uses explanations explicitly towards prediction, while the self-rationalization setting falls short. Hase and Bansal (2022) argues that explanations are most suitable as input for predicting, and Kumar and Talukdar (2020) designed a system to generate label-wise explanations, which is aligned with our design hypothesis. Nevertheless, there exist other works (Wiegreffe et al., 2021; Marasovic et al., 2022; Zelikman et al., 2022) that explore the use of self-rationalization setting. We include the self-rationalization setting in our human

evaluation of the explanation quality in Section 4.4.

## 3 Dual-Model AL System

### 3.1 System Architecture

Figure 1 illustrates our proposed dual-model AL framework. The system comprises three primary modules: 1) **an explanation-generation model** that takes the data, fine-tunes on human-annotated explanations, and generates free-form explanations; 2) **a prediction model** that accepts the data content and the generated explanations as input, fine-tunes on human-provided labels, and predicts the final label; 3) **an AL data selector** that selects a set of representative examples in terms of the semantic similarity between each unlabeled data text and labeled data's human explanations. The AL data selector plays a crucial role in finding a small, highly representative set of samples at every iteration, and further details of our AL selector are in Section 3.2.

In each AL iteration, after the data selector samples unlabeled examples for human annotations, we first fine-tune the explanation-generation model supervised by human-provided free-form explanations. Then, we instruct this model to generate explanations for the same set of data. Subsequently, we fine-tune a prediction model using the data content and explanations generated by the previous model as input, supervised by human-annotated labels. The fine-tuning process teaches the prediction model to rely on the explanations for predictions (Yao et al., 2023). Additionally, we fine-tune the prediction model with model-generated explanations instead of human-annotated ones for better alignment during inference, especially when no human annotations are available. After each AL iteration, we evaluate the framework on a standalone evaluation data split.

Both the explanation-generation model and the prediction model can be any SoTA sequence-to-sequence models, such as BART (Lewis et al., 2020) and T5 (Raffel et al., 2020). In this work, we utilize T5 as the backbone for both models and design a prompt-based input template for both models, as shown in Table 1, inspired by a few existing works (Schick and Schütze, 2021; Gao et al., 2021; Zhou et al., 2023). To elucidate how each prompt addresses a different part of data content:

1) "*explain:*" and "*question:*" are the leading prompts in the explanation-generation model and the prediction model, respectively, indicating different tasks for both models and are followed by

---

**Explanation-generation Model:**
**Training Input** **explain:** what is the relationship between *[hypothesis]* and *[premise]* **choice1:** entailment **choice2:** neutral **choice3:** contradiction
**Training Target** *[human annotated explanations]*
**Model Generation** *[generated free-form explanation]*

---

**Prediction Model:**
**Training Input** **question:** what is the relationship between *[hypothesis]* and *[premise]* **choice1:** entailment **choice2:** neutral **choice3:** contradiction **<sep> because** *[generated free-form explanation]*
**Training Target** *[human annotated label]*
**Model Prediction** *[predicted category]*

---

Table 1: The prompt-based input templates for both models in our system, with the e-SNLI (Camburu et al., 2018) dataset as an example.

the original task content. For the e-SNLI dataset, the task content becomes "what is the relationship between" the hypothesis and premise sentences;

2) "*choiceN*" is followed by candidate answers, where $N \in [1, 3]$ for the e-SNLI dataset corresponds to entailment, neutral, and contradiction. We pass the choices to the explanation-generation model, expecting that it will learn to generate free-text explanations that may reflect potential relationships between the data content and the task;

3) for the prediction model, an additional prompt "*because:*" is followed by the explanations generated by the explanation-generation model. We use a special token to separate the original task content and the explanation."

### 3.2 AL Data Selector

---

**Algorithm 1** Our Data Diversity-based AL Selector

**Variables:**
$D_{train} \Rightarrow$ unlabeled data in train split
$D_{prev} \Rightarrow$ previously-annotated data
$d_p^{data} \Rightarrow$ data content as a string of $d_p$ (for e-SNLI, it is the premise and hypothesis
$d_p^{exp} \Rightarrow$ previously-annotated free-form explanation of $d_p$
$x \Rightarrow$ number of data to be selected each iteration
$n_{train} = len(D_{train}); \ n_{prev} = len(D_{prev})$
**for** $D_i \in D_{train}$ **do**
    **if** *iteration* == 0 **then**
        $score_{d_i} = \frac{1}{n_{train}} \cdot \sum_{d_p \in D_{train}} similarity(d_i^{data}, d_p^{data})$
    **else**
        $score_{d_i} = \frac{1}{n_{prev}} \cdot \sum_{d_p \in D_{prev}} similarity(d_i^{data}, d_p^{exp})$
    **end if**
**end for**
$D'_{train} = rank \ D_{train} \ by \ score$
$D_{selected} = \ select \ x \ data \ from \ D'_{train} with \ equal \ intervals$
**Human annotation on** $D_{selected}$
$D_{train} - = D_{selected}; D_{prev} + = D_{selected}$

---

According to recent surveys of AL (Settles, 2009; Olsson, 2009; Fu et al., 2013; Schröder and Niekler, 2020; Ren et al., 2021), there are two primary approaches for AL data selection:

model probability-based and data diversity-based approaches. Model probability-based approaches, firstly, aim to select examples about which the models are least confident. These approaches involve conducting inference on unlabeled data at every iteration, which consumes more time and computing resources. Unlike data diversity-based approaches, they are not model-agnostic, which may affect the effectiveness of the sampling strategies depending on the model in use.

Secondly, data diversity-based approaches leverage various data features, such as data distribution and similarity, to select a representative set of examples from the candidate pool while maximizing diversity. This paper introduces a **data diversity-based** AL selection strategy that shares a concept similar to traditional data-based clustering strategy Nguyen and Smeulders (2004) and core-set strategy. However, our strategy differs from traditional strategies because ours incorporates human-annotated explanations for selection. More specifically, our data selector aims to choose examples that are representative of the unlabeled data pool in terms of average similarity to human-annotated explanations of all previously-labeled data while maximizing the diversity of newly-selected data.

We assume that **human-annotated explanations contribute significantly to the model's prediction and convey more information than the original data content alone**. These explanations can reveal underlying relationships between concepts in the data content and the relations between the data content and choices. For instance, in the e-SNLI dataset, the data content consists of the concatenation of hypothesis and premise sentences. Later, we construct a baseline selector in the AL simulation experiment (Sec. 4.2) with the same setup, except that it only compares the similarity between data content. Additionally, we include random baseline and probability-based baseline strategies. Our results demonstrate that using human-annotated explanations for data selection consistently leads to improved prediction performance compared to using data content alone.

Here we delve into the details of our data-based AL data selector (shown in Algorithm 1). For each unlabeled data instance, we use sentence-transformers (Reimers and Gurevych, 2019) to calculate the semantic similarity between its data content and every previously annotated explanation. Then, we take the averaged similarity scores for

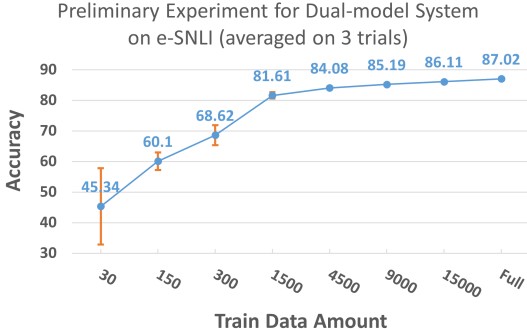

Figure 2: Preliminary experiment result of our dual-model system on e-SNLI (Camburu et al., 2018) dataset.

each unlabeled example and rank all the unlabeled data in terms of the average similarity score. To select the most representative data in the candidate pool while maximizing diversity, we choose examples from the ranked data list with equal intervals. Note that in the first iteration, since no previously annotated explanations are available, we compare the similarity between the data content.

## 4 Evaluation

We conduct the AL simulation experiment with the e-SNLI (Camburu et al., 2018) dataset. The primary objective is to justify that our proposed dual-model framework, when combined with human-annotated explanations in AL data selection, can effectively identify more representative and helpful data from a reasonably large-scale dataset.

Given that e-SNLI dataset comprises a substantial 549,367 examples in the train split, we performed a preliminary experiment to determine a reasonable number of candidate data for the AL simulation. This approach aims to save time and computing resources. Our goal is to identify an ideal candidate data size that would not introduce potentially biased feature distributions or significantly degrade model performance when compared to fine-tuning on the full dataset. We employ the pre-trained T5-base (Raffel et al., 2020) as the backbone for all the experiments and provide the hyperparameters in Appendix C.

### 4.1 Preliminary Experiment

The expected outcome of the aforementioned preliminary experiment is 1) to determine the upper bound of performance and observe how the performance of our dual-model system gradually decreases as we reduce the amount of training data, and 2) to identify a suitable candidate data size for

the AL simulation.

We also randomly sample the same amount of data for each category in the preliminary experiment to minimize potential bias introduced by uneven distribution, especially when the sampling size per iteration is very small. Specifically, we select eight different sampling amounts per category from the e-SNLI training split, ranging from [10, 50, 100, 500, 1500, 3000, 5000] and the complete data per category. Since the e-SNLI dataset consists of three categories: entailment, neutral, and contradiction, the total sampling size in each setting becomes [30, 150, 300, 1500, 4500, 9000, 15000, and 549, 367 (full train split)], respectively, as shown in Figure 2.

For each sampling setting, we conduct three trials to obtain an averaged result. In each trial, we fine-tune the explanation generation model and the prediction model once and conduct a hyperparameter search. The framework is then evaluated on the test split of e-SNLI (9, 824 examples).

The preliminary experiment results are shown in Figure 2, where the blue dot denotes the averaged prediction accuracy (in percentages) at each setting, and the red bar indicates the standard deviation of accuracy among three trials. Notably, with more than 1, 500 data per category, the performance drop compared to the full train split is inconspicuous (84.08% to 87.02%), while the standard deviation is below 0.5%. This observation indicates that using 1% of the original training data size only leads to a performance drop of merely 3%. Additionally, we found that even with only 10 data points per category (30 data in total), our system still achieves an average accuracy of 45%, although the deviation is relatively significant. Furthermore, when we extend the training data size from 100 to 500 data per category (300 to 1500 in total), a reasonably applicable setting in real-world scenarios, the accuracy can reach over 80% accuracy, showing promising results considering that the amount of training data is much smaller than the size of evaluation samples.

### 4.2 Simulation Experiment: Evaluation Setup

Based on the findings from the preliminary experiment, we decide to use 3, 000 examples per category (9, 000 in total) as the candidate unlabeled data pool for the Active Learning simulation.

Inspired by the few-shot evaluation guidance (Bragg et al., 2021), we conduct **80** trials for each AL setting and calculate the averaged per-

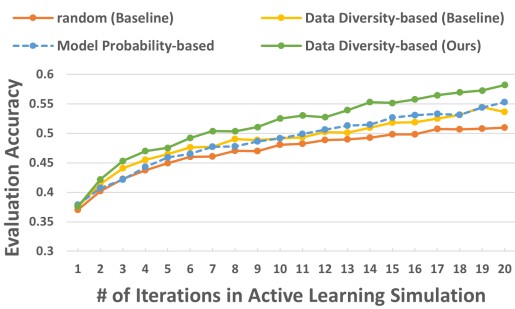

(a) Setting 1: 9 examples per iteration + 20 iterations

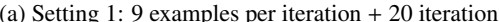
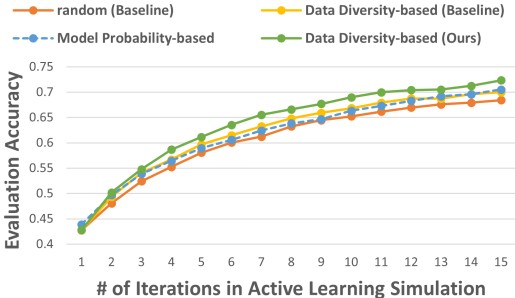

(b) Setting 2: 30 examples per iteration + 15 iterations

Figure 3: Results of AL Simulation experiment on our Dual-model system with different data selectors.

formance for ours and the baseline data selectors at every iteration. During each trial, we start by randomly selecting 3, 000 examples per category from the complete train dataset, then use the same data to conduct AL simulations with different data selectors in our dual-model framework. This way, we can ensure the performance differences during each trial are not due to different unlabeled data pools but to actual differences in the performance of the AL data selectors. For the evaluation, we randomly sample 300 examples per category (900 in total) from the test split of e-SNLI every trial and evaluate with the same test data after each iteration.

The AL simulation comprises two settings, where we simulate annotating 180 and 450 data instances, respectively. These two levels of data annotations reasonably mimic real-world scenarios where users have limited budgets, annotators, and data for annotation. Specifically, we experiment with the following two settings:

1) For every iteration, select **3** examples per category (9 in total) with **20** iterations, which results in **180** examples altogether;

2) For every iteration, select **10** examples per category (30 in total) with **15** iterations, which results in **450** examples altogether.

Our AL simulation experiment involves our data selector, two baselines, and an additional model

probability-based selector. Our data selector, described in Section 3.2, is a novel data diversity-based sampling strategy that leverages human-annotated explanations. For comparison, we use a random data selector as the basic benchmark and another traditional data diversity-based algorithm that shares the same procedures with ours, except that it only compares the similarity between each unlabeled data's content and the previously-labeled data's content, not using the human-annotated explanations. The probability-based selector conducts inference on unlabeled data and selects examples with the least probability at every iteration. We fix the same set of hyperparameters (Appendix C).

Worth noting that our data selector does not use task content in previously labeled examples; instead, we exclusively rely on human-annotated explanations to demonstrate their greater utility compared to task content. In the first iteration, both ours and the data diversity baseline perform identically because no previously annotated data is available.

## 4.3 Simulation Experiment: Result

The AL simulation results are presented in Figure 3. To explain the diagrams in detail, each dot is the average accuracy on 80 trials at every iteration for each data selector. The green/yellow/red/blue dots denote our data selector/data diversity-based baseline/random selector/model probability-based selector, respectively. We observe that our data selector consistently maintains an advantage over the traditional data-based sampling baseline, while the traditional one consistently beats the random baseline by a significant margin. Additionally, we observe that the model probability-based selector outperforms the random baseline in both settings.

To summarize, our data selector outperforms both baselines in every iteration for both AL settings, indicating that **using human-annotated explanations in the data selector with our dual-model AL framework is more beneficial than using the data content alone.** Even with only 180 and 450 data to be annotated in each setting, our system can achieve 55% and 72% accuracy on average, respectively. We anticipate that our experiment will reach a similar performance around 85% as shown in Figure 2 but converge much faster than the random selector if we continue the AL process.

## 4.4 Human Evaluation Setup and Results

To qualitatively evaluate the explainability of the generated explanations from our system against

| Yes / No Count | Label | Exp. | Exp. → Label | Trustworthy AI |
|---|---|---|---|---|
| Ground-truth | 83 / 7 | 86 / 4 | 87 / 3 | 78 / 12 |
| Dual-model (ours) | 64 / 26 | 68 / 22 | 48 / 42 | 35 / 55 |
| Self-rationalization | 42 / 48 | 67 / 23 | 51 / 39 | 21 / 69 |

Table 2: Human evaluation results.

a SoTA few-shot explanation-generation system, the self-rationalization baseline (Marasovic et al., 2022), and the human ground-truth, we recruited three human participants to conduct a human evaluation following the prior literature (Xu et al., 2022). The self-rationalization baseline is a T5-base model, which uses the same input template of our explanation-generation model shown in Table 1 but asks the model to generate both the label and explanation simultaneously.

We leverage AL setting 1 described in Section 4.2 to fine-tune our system with a total of 180 examples over 20 epochs and use the same 180 examples to fine-tune the self-rationalization baseline. Both systems are used to infer the complete test split of e-SNLI after fine-tuning; then, we randomly sample 80 examples for the human study.

For each data instance, the rater is presented with the textual content of the *premise* and *hypothesis* of the original data paired with three sets of *labels* and *explanations* from our system, baseline system, and the human-annotated ground-truth from the e-SNLI dataset. Participants who are not aware of the source of each label-explanation pair are asked to answer four questions with [Yes/No]:

1) Is the Prediction correct?

2) Is the Explanation itself a correct statement?

3) Regardless of whether the AI Prediction and Explanation is correct or not, can the Explanation help you to understand why AI has such Prediction?

4) Will you trust & use this AI in real-world decision-making?

To ensure inter-coder consistency, we first conduct a 30-min tutorial session to educate all three participants with ten examples to build a consensus among them. In the actual experiment, each of the three participants is then asked to rate 30 data instances (20 unique ones and 10 shared ones), which make up a total of 70 data instances, and 360 ratings (3 rater*30 instances*4 questions). We first calculated the Inter-Rater Reliability score (IRR) among them for each of the four questions. With the IRR score of (Q1: 1, Q2: 0.89, Q3: 0.98, Q4: 0.87), we are confident that the three coders have the same criteria for further result analysis.

Our questions all have binary responses, and we

rely on Chi-square analysis (Elliott and Woodward, 2007) to examine the statistical significance of the rating groups' differences. As shown in Table 2, the participants rated human ground-truth explanations highest across all four dimensions. Between our system and the few-shot self-rationalization system (baseline), participants believe our systems' predicted labels are more likely to be correct, with 64 'valid' ratings out of 90 for our system versus 42 out of 90 ratings for the baseline. Chi-square test indicates such a difference is statistically significant ($\chi^2(1) = 21.61, p < 0.01$).

When asked whether they would trust the AI if there were such AI systems supporting their real-world decision-making, 35 out of 90 answered 'Yes' for our system, and it is significantly better than the baseline system (21 'Yes' out of 90) ($\chi^2(1) = 12.17, p < 0.01$). In comparison, 78 out of 90 times people voted that they would trust the human-annotated explanation's quality.

As for Question 2 ("the validity of the generated explanation") and Question 3 ("whether the generated explanation is supporting its prediction"), the human evaluation fails to suggest statistically meaningful results between our system and the baseline system ($\chi^2(1) = 0.06, p = 0.89$ for explanation validity, and $\chi^2(1) = 0.41, p = 0.52$ for explanation supporting prediction). In summary, human participants believe our system can outperform the baseline system on the label prediction's quality and the trustworthiness of AI dimensions. Still, there is a large space to improve as human evaluators believe the ground-truth label and explanation quality is much better than either AI system.

### 4.5 Ablation Study 1: Transfer to Multi-NLI

We conduct an ablation study with transfer learning through AL simulation from e-SNLI to Multi-NLI (Williams et al., 2018). This study explores whether the explanation-generation model trained on e-SNLI is helpful for AL on a similar task.

The transfer-learning ablation study consists of the following steps: 1) fine-tune an explanation-generation model using our AL framework on the e-SNLI dataset; 2) freeze the explanation-generation model and use it to generate explanations in the AL simulation for Multi-NLI; 3) fine-tune the prediction model for Multi-NLI at every iteration. Unlike the e-SNLI experiment, our AL data selection algorithm will use model-generated explanations to select examples at every iteration in the transfer learn-

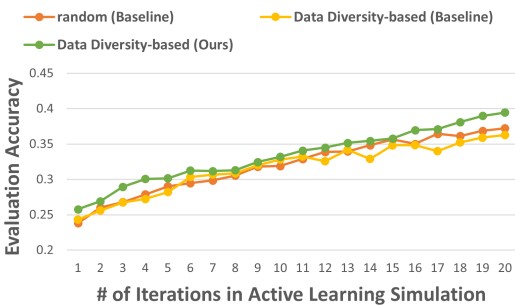

(a) Setting 1: 9 examples per iteration + 20 iterations

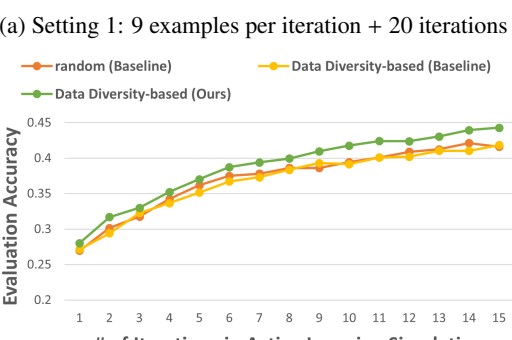

(b) Setting 2: 30 examples per iteration + 15 iterations

Figure 4: Ablation study results of AL simulation experiment on our Dual-model system with different data selectors on ECQA dataset.

ing AL simulation. We fine-tune the explanation-generation models on e-SNLI with the same two settings in the previous experiment, average the result on 15 trials of experiments, and keep consistent with every other hyper-parameters.

The ablation results are shown in Figure 6 of Appendix B. The blue/red lines denote the explanation-generation model is fine-tuned on e-SNLI with each setting in Section 4.2 correspondingly. We observe that the explanation-generation model consistently provided helpful explanations, leading to an improvement in the system's prediction performance, with accuracy reaching more than 65%. In addition, the explanation-generation model fine-tuned on more data can perform better, suggesting that it had learned to generate more helpful explanations.

### 4.6 Ablation Study 2: Our AL Framework on ECQA

We additionally conduct an AL Simulation experiment on ECQA (Aggarwal et al., 2021) (a recent dataset extends the CommonsenseQA dataset with high-quality human-annotated explanations) with our data selector, random baseline, and similarity-based baseline that does not use explanations. We comply with the same experiment settings for the e-SNLI AL simulations described in Section 4.2. The

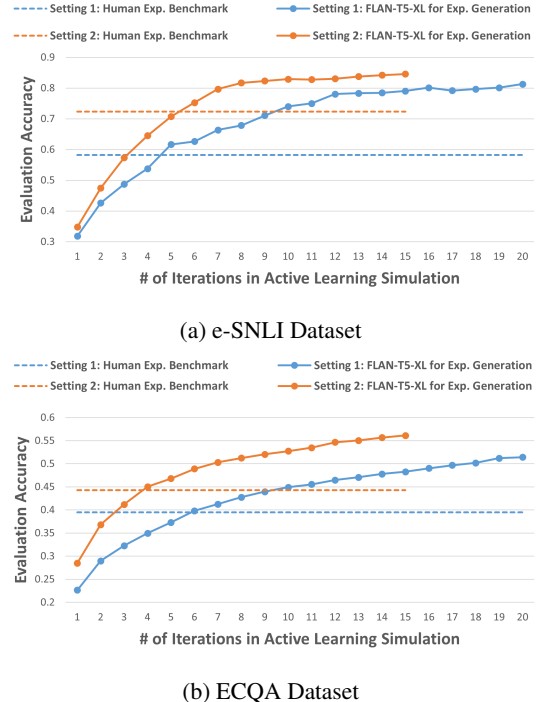

(a) e-SNLI Dataset

(b) ECQA Dataset

Figure 5: Ablation study results of AL simulation experiment with FLAN-T5-XL for explanation (exp.) generation in our Dual-model framework compared with best human-annotated explanations on e-SNLI (top) and ECQA (bottom) datasets.

results are shown in Figure 4, where our proposed data selection strategy can consistently outperform both baselines in both simulation settings. Interestingly, the similarity-based baseline performs similarly to the random baseline, which could be because using data content alone is not sufficient to select more helpful and representative examples while using human-annotated explanations can facilitate better data selection consistently.

### 4.7 Ablation Study 3: LLM for Explanation Generation

The recent prevalence of instructional-finetuned large language models (LLMs) (Wei et al., 2021; Chowdhery et al., 2022; Ouyang et al., 2022) with exceptional generation capabilities off-the-shelf enabled a straightforward idea upon our dual-model framework: **can LLMs generate natural language explanations that are on par or even of higher quality than human-annotated ones**, to facilitate the prediction model fine-tuning process? We conduct ablation experiments to leverage FLAN-T5-XL (Chung et al., 2022) for explanation generation in our framework to substitute the T5 model fine-tuned on human explanations (LLM-AL, hereinafter). We conduct the AL simulations

on e-SNLI and ECQA datasets to explore whether we can further reduce human annotation efforts.

The results are presented in Figure 5, where a horizontal dotted line represents the benchmark of the explanation generation model fine-tuned on human-annotated explanations in Section 4.3 and 4.6. The LLM-AL framework significantly outperforms the explanation generation model guided by human annotation in both Active Learning settings. However, **we hypothesize the LLM's explanation generation capability can vary from task to task**. It may be highly efficient in relatively easy tasks, such as e-SNLI and ECQA datasets, both of which are training datasets for FLAN-T5. Yet, LLMs may struggle to provide helpful explanations in complex real-world domain-specific tasks, where human experts' feedback may still be necessary and preferred. This leads to another potential avenue for future work: exploring the capability and limitations of leveraging LLMs for explanation generation in real-world scenarios.

## 5 Conclusion and Future Work

In summary, this paper introduces a novel dual-model AL system designed to address the common real-world need for domain experts to provide both classification labels and natural language explanations. Our system comprises a purpose-built data diversity-based AL example selector and two sequence-to-sequence language models, one for explanation generation and the other for label prediction. Through an AL simulation evaluation and a human assessment of the e-SNLI dataset, our results demonstrate the effectiveness of explanations in AL sampling with our system. They consistently outperform both baselines, and the explanations generated by our system are preferred over a state-of-the-art explanation-generation system.

Our work lays a step-stone towards a human-centered interactive AI solution (it can be easily implemented as an interactive system as illustrated in Fig 7 in Appendix D) that supports domain experts for their data annotation tasks. Many real-world tasks still require domain experts to review and annotate each data instance with a decision and an explanation for accountability purposes (e.g., a lawyer reviewing and signing off on a legal document). We invite fellow researchers to join us in advancing this research direction, essential for supporting this prevalent real-world requirement.

## 6   Limitations

In this paper, we demonstrate the effectiveness of our framework on a representative large-scale classification dataset (e-SNLI), but there are many other NLP tasks, such as question answering and commonsense reasoning. The generalizability of our system on other NLP tasks remains unexplored. Another limitation is that this work proposed a data diversity-based AL selector design. We benchmark it with a traditional data diversity-based selector as well as a model probability-based design to demonstrate the usefulness of explanations. Prior literature has proposed other designs, such as ensemble approaches, which are not evaluated in this paper.

## Acknowledgements

This work was supported by the Rensselaer-IBM AI Research Collaboration (http://airc.rpi.edu), which is part of the IBM AI Horizons Network (http://ibm.biz/AIHorizons).

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

# Appendix

## A  e-SNLI Examples

Table 3 illustrates an example data of each category in the e-SNLI dataset. Every data instance contains a premise and hypothesis along with a human annotated label and free-form explanation.

---

**Premise:** *This church choir sings to the masses as they sing joyous songs from the book at a church.*
**Hypothesis:** *The church is filled with song.*
**Label:** *entailment*
**Human-annotated explanation:** *"Filled with song" is a rephrasing of the "choir sings to the masses.*

---

**Premise:** *A man playing an electric guitar on stage.*
**Hypothesis:** *A man is performing for cash.*
**Label:** *neutral*
**Human-annotated explanation:** *It is unknown if the man is performing for cash.*

---

**Premise:** *A couple walk hand in hand down a street.*
**Hypothesis:** *A couple is sitting on a bench.*
**Label:** *contradiction*
**Human-annotated explanation:** *The couple cannot be walking and sitting a the same time.*

---

Table 3: Sample data of each category in e-SNLI (Camburu et al., 2018) dataset.

## B  Transfer Learning Ablation Study Diagrams

Figure 6 shows the results of our Ablation Study results described in Section 4.5. The explanation-generation model is fine-tuned from AL on e-SNLI dataset with two different AL settings, then we freeze the explanation-generation model to train the prediction model in AL simulation for Multi-NLI dataset under two settings. Setting 1/2 refers to the settings for Active Learning Simulation in Section 4.2.

## C  System Environment and Hyper-Parameters

The computing resource of all the experiments we conducted in this paper has 128 Gigabytes of RAM. In addition, we use 2 NVIDIA Tesla V100 GPU for the preliminary experiment and 8 NVIDIA Tesla V100 GPU for the AL simulation experiment.

### C.1  Preliminary Experiment

For the Preliminary experiment described in Section 4.1, we leverage the same set of fine-tuning hyper-parameters other than the number of fine-tuning epochs for the explanation-generation model (denotes as $M_{EG}$) and the prediction model (denotes as $M_P$). The same set

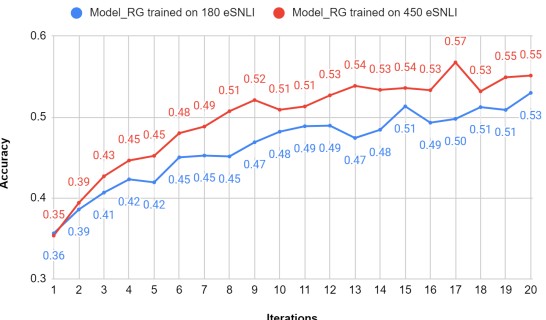

(a) Active Learning on Mulit-NLI using explanation-generation model from e-SNLI with Setting 1

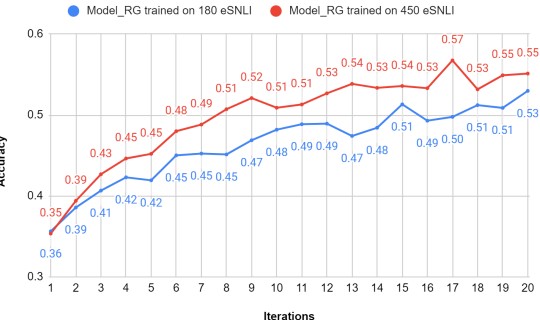

(b) Active Learning on Mulit-NLI using explanation-generation model from e-SNLI with Setting 2

Figure 6: Results of Transfer Learning Ablation Study of AL Simulation experiment on our Dual-model system from e-SNLI to Multi-NLI. Setting 1/2 refers to the settings for Active Learning Simulation in Section 4.2.

of hyper-parameters is: $batch\_size\_per\_GPU = 2; learning\_rate = 1e^{-4}; input\_max\_length = 512; target\_max\_length = 64$

We conduct a hyper-parameter search for the number of fine-tuning epochs for each amount of sampled examples, details are shown in Table 4.

| # of train data per category / total | epoch for $M_{RG}$ | epoch for $M_P$ |
|---|---|---|
| 10 / 30 | 25 | 100 |
| 50 / 150 | 25 | 250 |
| 100 / 300 | 10 | 250 |
| 500 / 1500 | 5 | 50 |
| 1500 / 4500 | 5 | 50 |
| 3000 / 9000 | 5 | 25 |
| 5000 / 15000 | 5 | 25 |
| Full | 1 | 1 |

Table 4: Fine-tuning epochs of each model in our dual-model system with different data amount settings.

### C.2  AL Simulation Experiment

For both of the AL Simulation settings we experimented in Section 4.2, we leverage the same set of hyper-parameters for fine-tuning our dual-model AL system: $batch\_size\_per\_GPU =$

$2; learning\_rate = 1e^{-4}; M_{EG}\_train\_epoch = 20, M_P\_train\_epoch = 250; input\_max\_length = 512; target\_max\_length = 64$

## D    Proposal for an Interactive System

Our proposed dual-model system can be easily implemented as an interactive human-centered AI system for supporting domain experts and human annotators in labeling both labels and explanations.

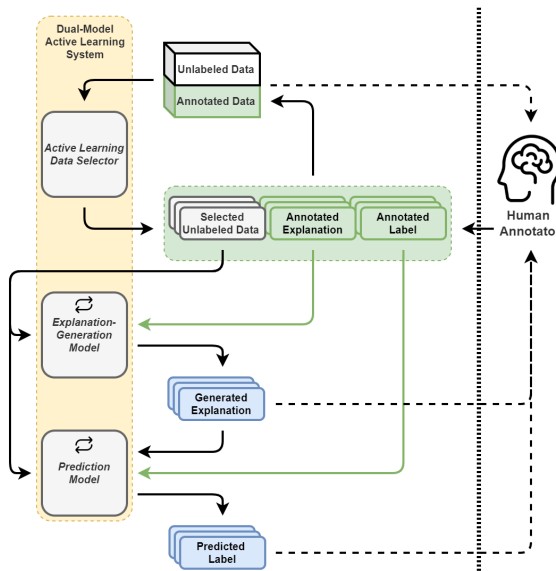

Figure 7: Our proposed dual-model system can be implemented as an interactive AL-based data annotation system to speed up users' annotation productivity. Such a system can simply have an interface with four output functions (i.e., display unlabeled data, display AL selected data, display generated-explanation, and display predicted labeled) and one input function (i.e., annotate label and explanation for the unlabeled data.