# OpenReview forum: "Beyond Labels: Empowering Human Annotators with Natural Language Explanations through a Novel Active-Learning Architecture"
_EMNLP/2023/Conference — EMNLP 2023 Findings_

### Official Review · Reviewer_DYu3 · 2023-07-31

**Soundness:** 4

**Excitement:**

4: Strong: This paper deepens the understanding of some phenomenon or lowers the barriers to an existing research direction.

**Justification For Ethical Concerns:**

I do not see ethical concerns in this paper.

**Missing References:**

N/A

**Paper Topic And Main Contributions:**

This paper considers the low-resource annotation setting, and proposes a novel active learning (AL) architecture to support experts who need to annotate both labels and explanations in low-resource scenarios.

The dual-model AL system consists of an explanation generation model and a prediction model. This system first collects a few human-provided labels and explanations. The explanations are used to fine-tune the generation model, which generates the explanations to fine-tune the prediction model, together with the human-provided labels. The prediction model finally predicts the label.

This paper conducts two AL simulations on a large-scale NLI dataset (eSNLI), and finds that the AL system consistently outperforms the baselines in two settings. Additionally, this paper recruits humans to eacluate the validity, explainability, and preference of the generated explanations. The explanations generated by the AL system is not as good as the human explanations, but are preferred compared to those generated by other baseline systems.

**Questions For The Authors:**

N/A

**Reasons To Accept:**

- The use of AL in the annotation of explanation and label is an important application.
- The proposed AL framework outperforms the baselines.
- Additionally, this paper further stuides the efficacy of the explanations generated by the AL system.
- This paper mentions the related literature comprehensively.

**Reasons To Reject:**

The AL setting is only verified on the NLI problems. It's unknown whether it can generalize further to other types of problems. Note that this paper has already included an experiment that transfers from e-SNLI to MNLI.

**Reproducibility:**

3: Could reproduce the results with some difficulty. The settings of parameters are underspecified or subjectively determined; the training/evaluation data are not widely available.

**Reviewer Confidence:**

4: Quite sure. I tried to check the important points carefully. It's unlikely, though conceivable, that I missed something that should affect my ratings.

**Typos Grammar Style And Presentation Improvements:**

- Figure 2: the standard deviation is usually plotted as error bars or regions, rather than a standalone line on the figure.

---

> ### Author Rebuttal · Authors · 2023-08-28
>
> We feel encouraged that you recognize our paper’s contribution to the NLP community, specifically for the active learning techniques, and sincerely thank the positive feedback. All concerns raised will be fully addressed in the camera-ready version, such as adding results, improving the presentation for plots and narratives, and fixing typos.
>
>
> > ”The AL setting is only verified on the NLI problems. It's unknown whether it can generalize further to other types of problems. Note that this paper has already included an experiment that transfers from e-SNLI to MNLI.”
>
> We hypothesize that natural language explanations contain more helpful and informative content compared with the data content alone, as mentioned in Section 3.2. As a result, our similarity-based selection strategy can distinguish those instances that may look similar (e.g., have a close similarity score by comparing data content alone) but need different rationales to solve, where the traditional strategy falls short. We justified this hypothesis on e-SNLI and the effectiveness of our framework for transfer learning toward Multi-NLI.
>
> We would like to point out that there exist very few datasets with high-quality human-annotated natural language explanations, and some of the existing datasets have been criticized for low-quality human explanation annotations, such as CoS-E and ComVE, so the Active Learning experiment on these datasets will not be faithful or reliable.
>
>
> Additionally, we conduct the Active Learning SImulation experiment on ECQA (a recent dataset extends CommonsenseQA dataset with high-quality human-annotated explanations) with our data selector, random baseline and similarity-based baseline that does not use explanations. The results are shown below (we will convert it to the diagram and add to the paper), where our proposed data selection strategy can consistently outperform both baselines in both simulation settings. Interestingly the similarity-based baseline perform similarly as the random baseline, which could because using data content alone is not sufficient to select more helpful and representative examples, while using human-annotated explanations can facilitate better data selection consistently.
>
> 30 examples per iteration and 15 iterations altogether
> |Data Selection Strategy|1|2|3|4|5|6|7|8|9|10|11|12|13|14|15|
> |--------|--|--|--|--|--|--|--|--|--|--|--|--|--|--|--|
> |Random|0.26|0.30|0.31|0.34|0.36|0.37|0.37|0.38|0.38|0.39|0.40|0.40|0.41|0.42|0.41|
> |Similarity Baseline|0.27|0.29|0.32|0.33|0.35|0.36|0.37|0.38|0.39|0.39|0.40|0.40|0.41|0.41|0.41|
> |Exp. Similarity (ours)|**0.28**|**0.31**|**0.33**|**0.35**|**0.37**|**0.38**|**0.39**|**0.39**|**0.40**|**0.41**|**0.42**|v0.42**|**0.43**|**0.43**|**0.44**|
>
>
> 9 examples per iteration and 20 iterations altogether
> |Data Selection Strategy|1|2|3|4|5|6|7|8|9|10|11|12|13|14|15|16|17|18|19|20|
> |--------|--|--|--|--|--|--|--|--|--|--|--|--|--|--|--|--|--|--|--|--|
> |Random|0.23|0.25|0.26|0.27|0.28|0.29|0.30|0.30|0.31|0.32|0.33|0.34|0.34|0.35|0.35|0.35|0.36|0.36|0.37|0.37|
> |Similarity Baseline|0.24|0.25|0.26|0.27|0.28|0.30|0.31|0.31|0.32|0.33|0.33|0.33|0.34|0.33|0.35|0.35|0.34|0.35|0.36|0.36|
> |Exp. Similarity (ours)|**0.25**|**0.26**|**0.28**|**0.30**|**0.30**|**0.31**|**0.31**|**0.31**|**0.32**|**0.33**|**0.34**|**0.34**|**0.35**|**0.35**|**0.35**|**0.36**|**0.37**|**0.38**|**0.38**|**0.39**|
>
>
> We believe this approach can generalize to other datasets because the abovementioned hypothesis and additional results. In addition, our proposed data selection strategy should be inherently both model-agnostic and task-agnostic.
>
>
> > “Figure 2: the standard deviation is usually plotted as error bars or regions, rather than a standalone line on the figure.”
>
> Thank you very much for the suggestion. We will adjust the standard deviation of Figure 2 into error bars in the camera-ready version if this paper gets accepted.

---

### Official Review · Reviewer_BmAS · 2023-08-01

**Soundness:** 4

**Excitement:**

2: Mediocre: This paper makes marginal contributions (vs non-contemporaneous work), so I would rather not see it in the conference.

**Missing References:**

*Using templates for classification*

[Exploiting Cloze Questions for Few Shot Text Classification and Natural Language Inference](https://aclanthology.org/2021.eacl-main.20/)

[Making Pre-trained Language Models Better Few-shot Learners](https://aclanthology.org/2021.acl-long.295)

*Templated explanation generation for classification with explanations*

[FLamE: Few-shot Learning from Natural Language Explanations](https://aclanthology.org/2023.acl-long.372)

**Paper Topic And Main Contributions:**

This paper is about incorporating natural language explanations into an active learning framework.

The main contributions of the paper is combining a dual-model architecture (an explanation generator and a classifier) with a data-diversity-based active learning heuristic. The authors demonstrate the effectiveness of this approach on e-SNLI over baselines.

**Questions For The Authors:**

- Despite the experimental effectiveness, why does it make sense to compare similarities between data instances and natural language explanations directly? Do you expect this approach to generalize to other datasets?

**Reasons To Accept:**

- While the idea of generating natural language explanations to assist model prediction isn't new, using it in guiding AL data selection is creative.
- The experiment design is solid, and shows that the proposed diversity-based data selection strategy outperforms other baselines.

**Reasons To Reject:**

- The effectiveness of the approach is only shown on e-SNLI. The authors could use non-NLI datasets and human annotators to further demonstrate the generality of the method.
- Introducing natural language explanations into an active learning setup substantially increases the cost of data annotation, because annotators would use more time than when asked to provide labels alone. How would the proposed approach perform against a random selection baseline (without explanations) with equal budget for annotation?

**Reproducibility:**

4: Could mostly reproduce the results, but there may be some variation because of sample variance or minor variations in their interpretation of the protocol or method.

**Reviewer Confidence:**

3: Pretty sure, but there's a chance I missed something. Although I have a good feel for this area in general, I did not carefully check the paper's details, e.g., the math, experimental design, or novelty.

**Typos Grammar Style And Presentation Improvements:**

Overall I find the writing style too verbose, which makes the paper difficult to read.

---

> ### Author Rebuttal · Authors · 2023-08-28
>
> We feel encouraged that you recognize our paper’s contribution to the NLP community, specifically for the active learning techniques, and sincerely thank the constructive feedback. All concerns raised will be fully addressed in the camera-ready version, such as adding the suggested references, improving the presentation for plots and results, improving the writing style, and fixing typos.
>
> We would like to address every comment and question below:
>
> > “The effectiveness of the approach is only shown on e-SNLI. The authors could use non-NLI datasets and human annotators to further demonstrate the generality of the method.”
>
> We totally agree with you. However, there exist very few datasets with high-quality human-annotated natural language explanations, and some of the existing datasets have been criticized for low-quality human explanation annotations, such as CoS-E and ComVE.
> In this paper, we primarily conduct the experiment on NLI tasks, with e-SNLI and transfer-learning on Multi-NLI, so that the results are more faithful and reliable.
>
>
> Additionally, we conduct the Active Learning SImulation experiment on ECQA (a recent dataset extends CommonsenseQA dataset with high-quality human-annotated explanations) with our data selector, random baseline and similarity-based baseline that does not use explanations. The results are shown below (we will convert it to the diagram and add to the paper), where our proposed data selection strategy consistently outperform both baselines in both simulation settings. Interestingly the similarity-based baseline perform similarly as the random baseline, which could because using data content alone is not sufficient to select more helpful and representative examples, while using human-annotated explanations can facilitate better data selection consistently.
>
> 30 examples per iteration and 15 iterations altogether
> |Data Selection Strategy|1|2|3|4|5|6|7|8|9|10|11|12|13|14|15|
> |--------|--|--|--|--|--|--|--|--|--|--|--|--|--|--|--|
> |Random|0.26|0.30|0.31|0.34|0.36|0.37|0.37|0.38|0.38|0.39|0.40|0.40|0.41|0.42|0.41|
> |Similarity Baseline|0.27|0.29|0.32|0.33|0.35|0.36|0.37|0.38|0.39|0.39|0.40|0.40|0.41|0.41|0.41|
> |Exp. Similarity (ours)|**0.28**|**0.31**|**0.33**|**0.35**|**0.37**|**0.38**|**0.39**|**0.39**|**0.40**|**0.41**|**0.42**|v0.42**|**0.43**|**0.43**|**0.44**|
>
>
> 9 examples per iteration and 20 iterations altogether
> |Data Selection Strategy|1|2|3|4|5|6|7|8|9|10|11|12|13|14|15|16|17|18|19|20|
> |--------|--|--|--|--|--|--|--|--|--|--|--|--|--|--|--|--|--|--|--|--|
> |Random|0.23|0.25|0.26|0.27|0.28|0.29|0.30|0.30|0.31|0.32|0.33|0.34|0.34|0.35|0.35|0.35|0.36|0.36|0.37|0.37|
> |Similarity Baseline|0.24|0.25|0.26|0.27|0.28|0.30|0.31|0.31|0.32|0.33|0.33|0.33|0.34|0.33|0.35|0.35|0.34|0.35|0.36|0.36|
> |Exp. Similarity (ours)|**0.25**|**0.26**|**0.28**|**0.30**|**0.30**|**0.31**|**0.31**|**0.31**|**0.32**|**0.33**|**0.34**|**0.34**|**0.35**|**0.35**|**0.35**|**0.36**|**0.37**|**0.38**|**0.38**|**0.39**|
>
> We believe this approach can generalize to other datasets because the abovementioned hypothesis and additional results. In addition, our proposed data selection strategy should be inherently both model-agnostic and task-agnostic.
>
> We also plan to extend our framework to other NLP tasks and specific real-world tasks in the future, such as recruiting human experts for explanation annotation in specialized domains and experimenting with our proposed methodology.
>
>
> > ”Introducing natural language explanations into an active learning setup substantially increases the cost of data annotation, because annotators would use more time than when asked to provide labels alone. How would the proposed approach perform against a random selection baseline (without explanations) with equal budget for annotation?”
>
> We agree this is an important and realistic topic about the tradeoff between gathering more varieties of human feedback and reducing the annotation cost while talking about deploying an active learning framework in real-world scenarios. From our perspective, we believe there is no gold standard answer, and it highly depends on the quality of annotated feedback (e.g., natural language explanations) and tasks. We believe the cost (time, labor, money, etc.) of additional annotation needed with our framework is reasonable and manageable with the help of active learning, where we are aiming to use as a minimum amount of annotated data as possible.
>
> We would also like to emphasize three important advantages of our proposed framework and collecting human-annotated explanations in real-world scenarios:
> 1. There exists plenty of real-world scenarios that need not only label annotations but also explanation annotations (so the cost of annotation is already paid and the explanations are necessary), such as doctors need to write down explanations for diagnosis and clinical decisions but today's label-only active learning frameworks don't support such scenarios and not make use of the explanation annotations.
> 2. By asking annotators to explicitly write down their reasoning process (e.g., explanations), they will form a clear chain of reasoning in their mind to verify the label annotation, so this could improve the quality of label annotations
> 3. Compared with traditional end-to-end training frameworks, our framework can provide faithful natural language explanations that explicitly contribute to model prediction and are easy for humans to understand and trust model predictions.
>
> In addition, we conduct experiments to leverage a freezed FLAN-T5-XL, state-of-the-art large language model (LLM) for explanation generation in the e-SNLI dataset, where we could further reduce the need for human explanations. The LLM-AL framework can further outperform the explanation generation model guided by human annotation in both Active Learning settings, we can include the result in this paper. However, the LLM’s explanation generation capability can vary from task to task, where it could be very efficient in relatively easy tasks, but may fail to provide helpful explanations in difficult domain-specific tasks, and human experts’ feedback may still be needed and preferred. It can lead to another line of future work to explore the capability and limitations of leveraging LLMs for explanation generation in real active learning scenarios, which does not fit into our paper.
>
>
> > “Despite the experimental effectiveness, why does it make sense to compare similarities between data instances and natural language explanations directly? Do you expect this approach to generalize to other datasets?”
>
> We hypothesize that natural language explanations contain more helpful and informative content compared with the data content alone, as mentioned in Section 3.2. As a result, our similarity-based selection strategy can distinguish those instances that may look similar (e.g., have a close similarity score by comparing data content alone) but need different rationales to solve, where the traditional strategy falls short. We justified this hypothesis on e-SNLI and the effectiveness of our framework for transfer learning toward Multi-NLI.
>
> We expect this approach can generalize to other datasets because the abovementioned hypothesis and the proposed data selection strategy should be both model-agnostic and task-agnostic.
>
>
> > “Overall I find the writing style too verbose, which makes the paper difficult to read.”
>
> We will revise the paper narrative to make our claims and statements precise and clear in the final camera-ready version, to avoid any potential misunderstanding and verbal feelings.
>
>
> > Missing References
>
> We are truly thankful that you brought up these related works, and we will add these papers [1][2][3] as references in the camera-ready version if this paper gets accepted.
>
> [1] Schick, Timo, and Hinrich Schütze. "Exploiting cloze questions for few shot text classification and natural language inference." arXiv preprint arXiv:2001.07676 (2020).
>
> [2] Gao, Tianyu, Adam Fisch, and Danqi Chen. "Making pre-trained language models better few-shot learners." arXiv preprint arXiv:2012.15723 (2020).
>
> [3] Zhou, Yangqiaoyu, Yiming Zhang, and Chenhao Tan. "FLamE: Few-shot Learning from Natural Language Explanations." arXiv preprint arXiv:2306.08042 (2023).

---

### Official Review · Reviewer_LfJH · 2023-08-11

**Soundness:** 2

**Excitement:**

3: Ambivalent: It has merits (e.g., it reports state-of-the-art results, the idea is nice), but there are key weaknesses (e.g., it describes incremental work), and it can significantly benefit from another round of revision. However, I won't object to accepting it if my co-reviewers champion it.

**Missing References:**

References related to multiple works done in Active Learning unlabeled data selection and labeling it to tune models are missing.
For example:
Incremental Learning with Unlabeled Data in the Wild.   Lee et. al(CVPR 2019)
Negative sampling in semi-supervised learning.       Chen et al.(ICML 2020)
Label Efficient Semi-Supervised Learning via Graph Filtering.       Li et al.(CVPR 2019)

**Paper Topic And Main Contributions:**

The paper proposes a novel AL(Active Learning) architecture to support experts who need to annotate both labels and explanations in low-resource scenarios. The architecture incorporates an explanation-generation model that can utilize human-annotated explanations and generate such explanations. It uses publicly available T5 model to generate explanations and the labels correspondingly for the data.

**Questions For The Authors:**

1. The paper doesn't have a result table showing the comparison of different pre-trained models on the unlabeled data sampled.
2. Also, the paper doesn't have multiple unlabeled data selection techniques. How they performed.

**Reasons To Accept:**

The strengths of the paper are:
1. It proposes a dual model that is an explanation-generation model guided by human-provided explanations
2. a prediction model that takes the data text and explanations generated by the explanation generation model for prediction.
3. An AL(Active Learning) data selector that selects a set of representative examples in terms of semantic similarity between each unlabeled data text and labeled data’s human explanations.
4. The paper proposes a novel data diversity-based AL(Active Learning) data selection strategy to exploit the explanation annotation.

**Reasons To Reject:**

Reasons to reject:
1. Paper uses a simple way of selecting unlabeled data which is very similar to human-annotated data via calculating similarity with the help of sentence transformer embeddings.
2. Paper uses a pre-trained T5 base model fine-tuned on small human-curated data +unlabeled data(labeled via similarity as mentioned in 1) to generate explanations and the corresponding labels for the data. The defined technique doesn't seem to be novel.
3.  There are multiple data selection strategies from unlabeled data which the author hasn't compared
 eg: Incremental Learning with Unlabeled Data in the Wild.   Lee et. al(CVPR 2019)
Negative sampling in semi-supervised learning.       Chen et al.(ICML 2020)
Label Efficient Semi-Supervised Learning via Graph Filtering.       Li et al.(CVPR 2019)
4. Missing results table so as to see how different pre-trained models and data labelling techniques performed.

**Reproducibility:**

3: Could reproduce the results with some difficulty. The settings of parameters are underspecified or subjectively determined; the training/evaluation data are not widely available.

**Reviewer Confidence:**

5: Positive that my evaluation is correct. I read the paper very carefully and I am very familiar with related work.

---

> ### Author Rebuttal · Authors · 2023-08-28
>
> We sincerely thank your review and feedback on our paper. All concerns raised will be fully addressed in the camera-ready version, such as adding the suggested references, improving the presentation for results, improving the writing style, eliminating misunderstandings, and fixing typos.
>
> We would like to clarify a few misunderstandings regarding our contributions and novelty. The misunderstanding could possibly be due to our narrative, and we will revise the paper narrative to make our claims and statements precise and clear in the final camera-ready version:
>
> > “Paper uses a pre-trained T5 base model fine-tuned on small human-curated data +unlabeled data(labeled via similarity as mentioned in 1) to generate explanations and the corresponding labels for the data. The defined technique doesn't seem to be novel. “
> >
> > “Paper uses a simple way of selecting unlabeled data which is very similar to human-annotated data via calculating similarity with the help of sentence transformer embeddings.”
>
> One of our primary contributions is to **propose a novel Active Learning similarity-based data selector that leverages human-annotated explanations, and we justify the use of explanations to help better select representative and helpful examples**. The hypothesis behind our proposed Active Learning data selector is that we assume the human-annotated explanations can convey more helpful and richer information than the original data content alone, which can facilitate better selecting representative and helpful examples compared with using data content alone (highlighted in Section 3.2).
>
> We showed the importance of human-annotated explanations in selecting representative samples which was previously overlooked, and there were no techniques in the past that showed how to use these explanations in selecting the Active Learning examples. Here we showed one technique (a similarity-based strategy) that is simple but shows promising results on different NLP tasks.
>
> We justified our hypothesis through the Active Learning Simulation, where the similarity-based baseline follows the same similarity calculation algorithm but only uses data content. We additionally implement a model probability-based baseline selector because these two baselines represent the primary types of traditional data selectors. We justify the effectiveness and helpfulness of using natural language explanations in data selection because our proposed data selector can consistently outperform both baselines at every iteration in two active learning simulation settings (Figure 3).
>
> In addition, we are the first to propose this novel active learning framework, including the novel AL data selection strategy and the dual-model architecture, to the best of our knowledge.
> Our proposed framework provides a substantial contribution to the community that better incorporates human natural language feedback for enhancing data selection, model finetuning, and model explainability while reducing annotation effort.
>
> Based on our results and contributions, we justified our hypothesis that natural language explanations are indeed helpful in active learning data selection, and we broad up the research space of further exploring better generating and use of natural language explanation in active learning data selectors, which will be another full of work we are planning to explore in the future.
>
>
> > “The paper doesn't have a result table showing the comparison of different pre-trained models on the unlabeled data sampled. “
> >
> > “Also, the paper doesn't have multiple unlabeled data selection techniques. How they performed.”
>
> We appreciate you bringing up some related data selection strategies, and we will add these papers as references. To clarify, we are aware that there are a lot of different data selection strategies, and there does not exist one that is better than any others, because the effectiveness of different strategies can vary from task to task. In this paper, we include two different types of data selector baselines in our Active Learning simulation (data diversity-based and model probability-based), which are two primary types of traditional data selectors.
>
> In this work, one of our primary contributions is to justify the effectiveness of incorporating natural language explanations in active learning data selection, compared with traditional AL data selection strategies that do not take this type of human feedback into consideration. Given our hypothesis regarding natural language explanations above, our similarity-based selection strategy can distinguish those instances that may look similar (e.g., have a close similarity score by comparing data content alone) but need different rationales to solve, where the traditional strategy falls short. We justified our hypothesis through the AL simulation by comparing it with the traditional similarity-based strategy and the model probability-based strategy without explanations (plotted in Figure 3).
>
> Specifically, we focus on demonstrating the effectiveness of incorporating human-annotated natural language explanations into data selection with a data similarity-based strategy and compare with traditional data similarity-based and model probability-based strategy that does not use such human feedback. We are proposing a new data selector for a more real-world and new scenario, where human domain experts need to label both labels and explanations. We provide the rationales for focusing on the data diversity-based selection strategy in Section 3.2 that the model probability-based approaches highly depend on the models and can not faithfully justify the effectiveness of using natural language explanations, but the data similarity-based strategy is purely based on the data features and does not rely on model architecture inherently. Thus, the experiment across different models is less relevant to our proposed methodology, and we do not include such an experiment in this paper. It is worth mentioning that we did implement a probability-based strategy as a baseline to compare the effectiveness of our strategy.
>
> We agree it will be an interesting and important idea of future work to comprehensively compare a large number of existing active learning data selection strategies on a variety of different tasks, which will consist of a large amount of work alone and will not fit in our current work.
>
> Additionally, we conduct the Active Learning SImulation experiment on ECQA (a recent dataset extends CommonsenseQA dataset with high-quality human-annotated explanations) with our data selector, random baseline and similarity-based baseline that does not use explanations. The results are shown below (we will add the tables in the Appendix section to visualize the clear gain), where our proposed data selection strategy can consistently outperform both baselines in both simulation settings. Interestingly the similarity-based baseline perform similarly as the random baseline, which could because using data content alone is not sufficient to select more helpful and representative examples, while using human-annotated explanations can facilitate better data selection consistently.
>
> 30 examples per iteration and 15 iterations altogether
> |Data Selection Strategy|1|2|3|4|5|6|7|8|9|10|11|12|13|14|15|
> |--------|--|--|--|--|--|--|--|--|--|--|--|--|--|--|--|
> |Random|0.26|0.30|0.31|0.34|0.36|0.37|0.37|0.38|0.38|0.39|0.40|0.40|0.41|0.42|0.41|
> |Similarity Baseline|0.27|0.29|0.32|0.33|0.35|0.36|0.37|0.38|0.39|0.39|0.40|0.40|0.41|0.41|0.41|
> |Exp. Similarity (ours)|**0.28**|**0.31**|**0.33**|**0.35**|**0.37**|**0.38**|**0.39**|**0.39**|**0.40**|**0.41**|**0.42**|**0.42**|**0.43**|**0.43**|**0.44**|
>
>
> 9 examples per iteration and 20 iterations altogether
> |Data Selection Strategy|1|2|3|4|5|6|7|8|9|10|11|12|13|14|15|16|17|18|19|20|
> |--------|--|--|--|--|--|--|--|--|--|--|--|--|--|--|--|--|--|--|--|--|
> |Random|0.23|0.25|0.26|0.27|0.28|0.29|0.30|0.30|0.31|0.32|0.33|0.34|0.34|0.35|0.35|0.35|0.36|0.36|0.37|0.37|
> |Similarity Baseline|0.24|0.25|0.26|0.27|0.28|0.30|0.31|0.31|0.32|0.33|0.33|0.33|0.34|0.33|0.35|0.35|0.34|0.35|0.36|0.36|
> |Exp. Similarity (ours)|**0.25**|**0.26**|**0.28**|**0.30**|**0.30**|**0.31**|**0.31**|**0.31**|**0.32**|**0.33**|**0.34**|**0.34**|**0.35**|**0.35**|**0.35**|**0.36**|**0.37**|**0.38**|**0.38**|**0.39**|
>
> We believe this approach can generalize to other datasets because the abovementioned hypothesis and additional results. In addition, our proposed data selection strategy should be inherently both model-agnostic and task-agnostic.
>
> > Missing References
>
> We appreciate you bringing up these related data selection strategies, and we will add these papers [1][2][3] as references in the camera-ready version if this paper gets accepted.
>
> [1] Incremental Learning with Unlabeled Data in the Wild. Lee et. al(CVPR 2019)
>
> [2] Negative sampling in semi-supervised learning. Chen et al.(ICML 2020)
>
> [3] Label Efficient Semi-Supervised Learning via Graph Filtering. Li et al.(CVPR 2019)

---

### Meta-Review · Area_Chair_axg2 · 2023-09-18

**Recommendation:** 3

**Metareview:**

This paper proposes a new method to incorporate explanations from human annotators into active learning. The idea is a very reasonable extension to existing active learning approaches, and dual-model solution also seems clean and effective. There was some initial concern about the generalizability of the approach since the experiment only focused on NLI, which the authors justified by the lack of high-quality explanation annotation. During the discussion period, the authors conducted additional experiments on another dataset, which although showed smaller improvements compared to NLI, validates the soundness of the method. It would be exciting to see how the proposed approach complement alternative active learning strategies.

---

### Decision · Program_Chairs · 2023-10-07

**Decision:**

Accept-Findings

**Comment:**

This paper proposes a new method to incorporate explanations from human annotators into active learning. The idea is a very reasonable extension to existing active learning approaches, and dual-model solution also seems clean and effective. There was some initial concern about the generalizability of the approach since the experiment only focused on NLI, which the authors justified by the lack of high-quality explanation annotation. During the discussion period, the authors conducted additional experiments on another dataset, which although showed smaller improvements compared to NLI, validates the soundness of the method. It would be exciting to see how the proposed approach complement alternative active learning strategies.